# Variation of Salivary pH After Sweet and Oat Biscuit Intake—A Crossover Randomized Controlled Trial

**DOI:** 10.3390/foods14234141

**Published:** 2025-12-03

**Authors:** Cristina Teodora Preoteasa, Aura-Cristiana Bălțătescu, Anca Diana Cristea, Anca Axante

**Affiliations:** 1Department of Scientific Research Methodology and Ergonomics, Faculty of Dental Medicine, “Carol Davila” University of Medicine and Pharmacy, 041313 Bucharest, Romania; aura-cristiana.baltatescu0925@rez.umfcd.ro; 2Department of Endodontics, Faculty of Dental Medicine, “Carol Davila” University of Medicine and Pharmacy, 041313 Bucharest, Romania; diana.cristea@umfcd.ro

**Keywords:** food, saliva, dental caries, biomarkers, mastication

## Abstract

Aim: To assess salivary pH variation after consumption of two types of biscuits, i.e., cocoa biscuits with a sweet-creamy filling (Oreo original) and oat biscuits (Digestino oat). Method: A crossover randomized trial was conducted on a convenience sample of young adults with permanent dentition, whose diet included biscuits. Salivary pH was registered by pH strips, before and after biscuit intake, and then at 5 min intervals, over a period of 30 min. Results: Salivary pH had a similar pattern of variation for both biscuits used, but differences were noted, respectively, the range and maximum drop being slightly larger for Oreo original; timing of pH beginning to increase from maximum drop being delayed for Oreo original compared to Digestino oat, approximately 20 vs. 15 min after intake; the pH drop compared to baseline was not statistically significantly different at any timepoints for Digestino oat, but pH was statistically significantly lower starting from 15 to 25 min after intake for Oreo original. pH recovery at thirty minutes was frequent (69% of the participants) for Digestino oat, but it was rare (20% of the participants) for Oreo original. After the Oreo original intake, mouth rinsing with water enabled pH recovery (afterward in 60% of participants). Minimum salivary pH was strongly correlated to the initial pH for Oreo original intake (r = 0.780; *p* < 0.001), but moderately correlated for Digestino oat intake (r = 0.445; *p* < 0.012). Conclusions: Results suggest salivary pH registers similarities of the general pattern of variation after biscuit intake, but cocoa biscuits with sweet filling, compared to oat biscuits, seem to have a higher acidogenic effect.

## 1. Introduction

Mastication is an important oral function, and its optimal performance is related to oral health, which has an impact on the quality of lifewell-being [1]. Saliva plays an essential role in the optimal achievement of mastication, but also of digestion, both due to its components and due to the functions it performs [2].

According to the definition provided by a working group of the National Institutes of Health [3], a biomarker is an indicator that is objectively evaluated and measured in biological processes, whether physiological or pathological, or is used to quantify the pharmacological response to various therapeutic interventions [4]. Over time, researchers have sought to discover non-invasive ways to monitor the health of individuals or to detect the presence of certain molecules that indicate the existence of a disease. In recent decades, attention has been directed towards studies dedicated to the analysis of salivary biomarkers, saliva being considered an important source of information not only regarding the state of oral health, but also of general health as well, both for the current status and as an indicator of predictability for individual predisposition to certain diseases [5,6]. The consumption of different eatables is associated with changes in salivary biomarkers, which may be linked to oral health status and can have diagnostic and prognostic value.

Mastication, or the process of chewing, plays a fundamental role not only in the mechanical breakdown of food but also in maintaining oral and systemic health [7,8]. Mastication stimulates salivary secretion—a critical physiological process that contributes to the protection and balance of the oral environment. Saliva acts as a natural defense mechanism, buffering acids, facilitating remineralization of enamel, cleansing food debris, and maintaining microbial equilibrium within the oral cavity. As food enters the digestive tract through the mouth, it is first fragmented into smaller pieces and blended with saliva, allowing it to move more efficiently through the system. Chewing initiates the mechanical breakdown of food. Forming the bolus involves the processes of deformation and breakdown, supported by the coordinated activity of the teeth, the tongue, and salivary secretions. Saliva, composed mostly of water, also contains electrolytes, bicarbonate, mucin, and enzymes. It lubricates the food and begins dissolving it [9]. Chemical digestion in the mouth is limited, but it involves salivary amylase (ptyalin or alpha-amylase) and lingual lipase, both present in saliva. Salivary amylase, which is chemically identical to pancreatic amylase, breaks down starch into maltose and maltotriose, working best at a pH between 6.7 and 7.0. Another enzyme, present in the oral cavity, lingual lipase, hydrolyzes the ester bonds of triglycerides, producing diacylglycerols and monoacylglycerols [10].

Among the various salivary parameters, pH stands out as a key indicator of oral health status, reflecting the dynamic interplay between demineralization and remineralization processes that directly influence caries development and mucosal integrity. Quantifying changes in salivary pH following the mastication of different foods offers valuable insights into how dietary habits impact oral acidity, microbial activity, and overall dental health. Consequently, understanding and monitoring salivary pH variations serve as essential tools for identifying potential risks, guiding preventive strategies, and promoting optimal oral hygiene practices in both clinical and research contexts. So, the consumption of different eatables is associated with changes in oral/salivary biomarkers, which may be linked to oral health status and can have diagnostic and prognostic value.

Additionally, since oral health is closely linked to overall health, salivary pH analysis could also help explore connections with various systemic diseases. The variation in salivary pH is probably different in relation to the composition and other characteristics of the products consumed, so it is necessary to identify some factors that explain this and to explore the general trend manifested, considering the numerous variants of existing commercial products. Among the existing foods, biscuits were chosen because they are a snack quite frequently present in the dietary routine of the population of some geographical regions, such as our country. Research on salivary pH after biscuit intake is very limited, and very different in regard to method features, such as the type of study, sample features, biscuits consumed in regard to type and as commercial products, salivary pH evaluation, method of recording, and observation time and interval used [11,12,13,14]. From the previous referenced studies, only two were crossover studies [12,14], and none of them used random allocation of participants. The novel contribution of this research is more accurate knowledge of the pH curve after biscuit intake (data was recorded at fixed 5 min intervals for a period of 30 min after biscuit intake, which is more timepoints compared to previous studies), the differences observed between different types of biscuits, and data on pH recovery time after biscuit intake, including highlighting the beneficial effect of mouth rinsing with water. In addition, aspects of the outcomes measured (e.g., salivary pH range) and particularities of study design (e.g., randomized crossover trial) could be considered for shaping a standardized method more appropriate to control for error in future studies on this clinically relevant topic.

The main aim of this research is to assess salivary pH variation after the consumption of two types of biscuits, i.e., cocoa biscuits with a sweet filling (Oreo original) and oat biscuits (Digestino oat). The impact on the length of the follow-up and of mouth rinsing with plain water on salivary pH recovery to its baseline level was also assessed.

## 2. Materials and Methods

This research received approval from the Scientific Research Ethics Commission of the Carol Davila University of Medicine and Pharmacy, Bucharest (PO-35-F-03, No. 8367/2024).

A crossover randomized control trial was conducted. Participants were allocated to two groups, different in respect to the sequence of biscuits eaten, i.e., Oreo original (Mondelez International, Czech Republic) and Digestino oat (Tastino, Lidl, Germany). The generation of the allocation sequence was performed prior to beginning the trial. Block randomization, with a block size of 4, was used, i.e., each block included 2 participants from each of the 2 groups, resulting in a total of 6 possible combinations. Afterwards, the sequence corresponding to the random selection of blocks was generated, using the Microsoft Excel function RANDBETWEEN(1,6).

A convenience sample of participants was formed, with participants being recruited from November 2023 to March 2024, most of them being dental students from the Carol Davila University of Medicine and Pharmacy, Bucharest. Persons with ages ranging from 20 to 30 years who stated that they sometimes eat biscuits were included. Exclusion criteria included: known food allergy, known salivary gland dysfunction, oral pain or other acute symptoms with an impact on chewing function, and intake of drugs that may have an effect on salivary function. Prior to study inclusion, participants were informed about the study’s main features and signed an informed consent form, certifying their agreement to participate. Drawing from prior studies that have reported on this specific topic, salivary pH after biscuit intake, it was determined that a sample size of at least 10 participants is required for a parallel group trial [11,13]. Even so, after considering method differences and other previous crossover study reports on similar topics [15,16] regarding salivary pH variation after the intake of different eatables, a sample size of about 30 participants was targeted for this crossover trial.

The two commercial products of biscuits were used in this study: Oreo original (Mondelez International) and Digestino oat (Tastino, Lidl). These are two types of biscuits, i.e., Oreo original are cocoa biscuits with a sweet filling, and Digestino oat are oat biscuits. For 100 g, Oreo Original and Digestino oat have the following nutritional information: carbohydrates 72 g and 55.8 g; sugars 41 g and 12.5 g; and salt 0.44 g and 0.7g, respectively.

Data was collected through structured interviews for general aspects, and by the usage of pH strips for evaluating salivary pH.

The assessment of unstimulated salivary pH (Figure 1) was performed before biscuit intake and afterwards, immediately after, and for a 30 min follow-up period for each group. Participants were previously asked not to drink or eat 2 h prior to the assessment. According to the random allocation scheme previously generated, participants were asked to eat the biscuits in the corresponding sequence. After the registration of initial (resting) salivary pH, the participant was offered the first biscuit, according to their own sequence, and was instructed to chew bilaterally. The second measurement of the salivary pH was performed immediately after the participant swallowed the biscuit. Then, repeated measurements of salivary pH were performed at 5 min intervals, over a period of 30 min after biscuit intake. Afterwards, after registering the salivary pH for the first biscuit, the participant was asked to rinse the oral cavity with water, thus ensuring that the possible residues of the first biscuit were removed, and promoting pH recovery to its baseline [17]. According to our pilot assessments and previous studies [14,18], frequently in the 30 min following the consumption of eatables, the salivary pH returns to its baseline. Consequently, the wash-out period used was carried out 40 min after the first biscuit consumption, i.e., 30 min of observation after first biscuit intake, followed by a mouth rinse with plain water, followed by a ten-minute break. Afterwards, measurements related to the consumption of the second biscuit were conducted in the same manner as that for the first biscuit intake. Each participant was asked to attend a second meeting for the reassessment of the variation in salivary pH; this was conducted on a different day.

For registering the salivary pH, strips (Gemc Technology Group Ltd., Zhengzhou, China) with a measuring range from 4.5 to 9.0 units were used, with measurement intervals varying from 0.25 to 0.5 units, as observed in Figure 2. The same operator (A.C.B) performed all salivary pH recordings. Participants were asked to collect saliva into the mouth and, at the moments designated, to spit it into a disposable cup, in which the pH strip was dipped. After 15–30 s, the time required for color change, the value of the pH was registered by comparing the color of the strip to the color chart displayed on the packaging.

For the statistical analysis, IBM Statistics, version 22.0, was used. Considering that the data registered were not normally distributed, nonparametric tests were used for group comparison and correlation analysis, i.e., the Wilcoxon test, Friedman test, and Spearman test. The statistical significance threshold used was *p* < 0.05.

## 3. Results

Sample characteristics. In total, 31 participants were included, 23 were female and 8 male, with ages ranging from 20 to 29 years old, with a mean age of 24.3 years. Of these, 6 were smokers, and 10 reported frequently consuming carbonated drinks and/or citrus fruit juices. The following symptoms related to salivary function were reported as being found rather frequently: sensation of oral dryness (n = 2), sensation of dry lips (n = 18), and feeling thirsty (n = 14). Self-assessment of the saliva quantity was reported as either normal (n = 30) or excessive (n = 1), and the quality was reported as being either normal (n = 27) or thick (n = 4).

Participants were randomly allocated to two groups, different in respect to the sequence of the consumption of the two biscuit products: the first group (n = 15) ate the Oreo original first, then the Digestino oat, and the second group (n = 16) started with the Digestino oat, followed by the Oreo original.

Salivary pH variation related to biscuit consumption. The initial salivary pH, registered before the intake of Oreo original and Digestino oat, was not statistically significant different (Wilcoxon signed-rank test; *p* = 855). The variation in salivary pH registered a similar pattern for both biscuits used (Figure 3). Right after consumption, salivary pH increased about 0.7 units, decreasing afterwards to a value slightly above the initial pH, followed by its gradual decrease up until 15 to 20 min after consumption. This pattern was followed by an increase in the pH value until the end of the follow-up period, i.e., 30 min after consumption, with it reaching a value close to the initial pH. Even so, some differences were noted in regard to the variation in the salivary pH related to the consumption of these two types of biscuits. Compared to the consumption of the Digestino oat, the consumption of the Oreo original demonstrated a slightly higher increase in salivary pH, followed by a slightly lower decrease in salivary pH, and also the increase of salivary pH from the maximum drop started later for Oreo original than for Digestino oat. A non-statistically significant difference compared to the initial pH was noted after the 5 min until the 30 min follow-up for Digestino, but a statistically significant lower salivary pH than the initial pH was observed in the time frame corresponding to 15 to 25 min after consumption of Oreo biscuits (Table 1).

Compared to Digestino, the consumption of Oreo was related to a higher maximum pH, a lower minimum pH, and a wider pH range, but differences were not statistically significant (Table 2).

The initial value of salivary pH, registered before biscuit consumption, was positively correlated with both maximum and minimum pH for both biscuits. A strong correlation was noted between initial pH and minimum pH in the case of Oreo original biscuit consumption, and a moderate one for Digestino oat biscuits (Table 3).

Assessment of the length of the follow-up period and of the mouth rinse with plain water on the recovery of salivary pH to its baseline.For this assessment, only data related to the first biscuit intake was used. The time period considered consisted of the first 30 min (corresponding to follow-up after first biscuit intake) and the following 10 min (during which a mouth rinse with water was performed, then followed by a 10 min break).

Compared to baseline, the salivary pH was statistically significantly different at the 30 min follow-up, but not statistically significantly different at the 40 min follow-up (Table 4). Subgroup analysis revealed that only for the Oreo original was the value observed at the 30 min follow-up statistically significantly different from the initial value, but not different at the 40 min follow-up, suggesting the positive effect in this case on salivary pH recovery of water rinsing and an extra 10 min follow-up combined.

At the 30 min follow-up, pH recovered to baseline for 14 participants, 11 out of the 16 participants who consumed Digestino oat, and only 3 from the 15 participants who consumed Oreo original first, suggesting that 30 min after the consumption of Digestino oat, pH recovers to baseline in most cases, but this does not happen for Oreo original.

At the 40 min follow-up, a majority of the participants (n = 20) recovered their salivary pH to its baseline, composed of the same 11 from the group that consumed Digestino oat first, and 6 more (9 in total) from those who consumed Oreo original first. This suggests a positive effect of mouth rinse with water after Oreo consumption for the recovery of the salivary pH to its baseline.

Analyzing the pH variation from the 30 to 40 min follow-up (Table 5), the following were observed. In all participants in which salivary pH recovered at the 30 min follow-up, no other changes were noted afterwards at the 40 min follow-up. Regarding the rest of the participants, at the 40 min follow-up, no change was seen in 3 participants from the Digestino oat group, and changes occurred in 2 participants; out of the two, one presented a pH that became more acidic. At the 40 min follow-up, for the participants in the Oreo original group, no change was registered in 3 participants, and a change in terms of increasing pH towards its recovery was detected in 3 cases. These results suggest that water rinse had a positive effect on pH recovery, mainly after the consumption of the Oreo original, but not after the Digestino oat.

## 4. Discussion

The study results suggest that the variation in salivary pH has a similar pattern for the same eatables, meaning biscuits, but with variations between them in terms of range, maximum drop, and start of recovery, as seen by increasing its value. Cocoa biscuits with sweet filling, compared to oat biscuits, are associated with a slightly higher range, a lower maximum drop, and a delayed start point of recovery, as seen by the increasing of thepH value towards its baseline. In most cases, the salivary pH returns to its baseline 30 min after the consumption of oat biscuits. Salivary pH recovery was favored in the case of a mouth rinse with water after the consumption of the cocoa biscuits with sweet filling.

Snacking is a relatively frequently encountered habit [19,20,21] that was previously shown to increase the risk of caries [22], this being linked to different mechanisms, such as changes in salivary pH. Biscuits were chosen in this research, as they were previously found to be consumed rather frequently as snacks, in between meals [23], are abundant in carbohydrates [24], and considered to increase the risk of dental caries [25]. Biscuits are available in many variants, different in terms of composition and physical properties, with probable impacts, including changes in salivary pH and the risk of dental caries.

Pachori et al. [14] found a similar pattern of salivary pH variation after biscuit intake to that observed in our study. One aspect that was found in both studies, but different from the others on the topic [11,12,13], is the increase in salivary pH level immediately after consumption, an aspect found by Pachori et al. [14] for all solid foods analyzed, but for none of the liquid ones. We consider that this increase in salivary pH could be explained by food chewing acting as a stimulus for saliva production, as it is known that stimulated saliva has a higher pH than unstimulated saliva [26,27], explained by the increase flow rate and the increase in the concentration of bicarbonates and ion changes [28]. Afterwards, the pattern was similar in regard to salivary pH: on a general decreasing trend until the 15 min follow-up, and still being slightly above baseline at minute 5 after intake. One methodological difference is that, in other studies, pH values between minutes 15 and 30 were not recorded. At their endpoint, 30 min after intake, and similar to our results, pH was not statistically different from baseline. Even if the general pattern of salivary pH variation is very similar to the one presented in our research, the values of salivary pH are considerably different, having a lower range of variation, which may be linked to the differences between the studies, such as the biscuits used and participants’ features, i.e., children between 8 and 12 years in the study of Pachori et al. [14] and 20 to 29 years in ours.

Vivek Babu et al. [13] report on the variation in salivary pH after the intake of four different commercial products of biscuits, one of them being Oreo; the timepoints used were baseline, immediately after, and at 15 and 30 min after intake. According to their results, after biscuit intake, the lowest pH value was observed immediately after swallowing, and after the 30 min follow-up, a value that was below the baseline was still encountered, which was different in amount among the four commercial products used. Among the commercial products used, Oreo registered the lowest salivary pH level at all three timepoints after intake. The different results observed may be linked to the sample size used in this research (only 10 participants per group) and age difference (children of 10 to 15 years were included); it also bears mentioning that this was only a pilot study. Even so, their results suggest that when salivary pH variation after biscuit intake is assessed, a follow-up of at least 30 min is recommended. Also, their findings confirm that there are differences between the different types of biscuits.

Another study on the topic, reported by Kumar et al. [11], which compared different types of biscuits, found results that show the acidogenic effect of biscuits, which has a different level depending on biscuit type. In agreement with our results, oat biscuits had less acidogenic effect than that of cream or chocolate biscuits, and 15 min after intake (the follow-up time used in the study), the level of salivary pH was still lower than at baseline.

The studies that have reported on the topic of the variation in salivary pH related to biscuit consumption are very scarce, and they are also very different in terms of research method [11,12,13,14]. Firstly, they are not uniform regarding follow-up timepoints and total duration, which is important in order to know when the maximum pH drop occurs and how long it takes for the salivary pH to return to the baseline level (the recovery time). In the previously discussed studies, the follow-up varied between 12 and 30 min, and different timepoints were used, e.g., salivary pH from minute 15 to 30 after intake was not registered for either of the studies. These aspects regarding the method are important to note, considering our findings, specifically after Oreo original intake, the follow-up should be at least of 20 min after intake, since at that point the maximum drop of salivary pH was generally seen, and recommended to be at least 30 min, as until then the salivary pH was found to be statistically significantly different from baseline. Also, the commercial product used in the previously reported studies, and sometimes even the biscuit type used, was unclear [12]. This aspect poses, according to our results and others [11,13], differences in pH variation. These may explain the disagreement between some of the results mentioned above and suggest that more data is needed to know how to properly collect data on salivary pH, in order to acquire data on maximum drop level and recovery time; information that is important to know from a clinical point of view.

As suggested by our results, a simple procedure, particularly mouth rinsing with plain water, may have a positive effect on returning the salivary pH to its baseline value (contribute to pH recovery). This information is in accordance with previous studies, as the one reported by Singh et al. [17]. Snacking on high-carbohydrate foods, such as biscuits, can have a higher cariogenic potential due to food retention [29]. The positive effect of water rising on pH recovery could be related to removal of carbohydrate residue, but also too other mechanisms, such as the dilution of acids, the increase in salivary flow, or a combination of previous factors. If confirmed, this information presents both clinical and scientific research value. Shortening the period of recovery time of salivary pH after food intake could decrease the risk of dental caries; therefore, mouth rinsing with water after snacking could be recommended as a preventive behavior in daily life. Also, it could be used in crossover trials in which salivary pH is evaluated. If pH returned to baseline value, eatables could be administered on the same day; consequentially, this can pose some advantages as a lower risk of loss to follow-up in the case of crossover trials with data collected on different days, or testing differences within the same individual, and a lower sample size compared to two-arm trials.

Thus, saliva represents a mirror of the physiological processes that take place at every moment in the oral cavity, but also of the current state of one’s health, reflected by the biomarkers present at this level; at the same time, saliva also creates a picture of the individual predisposition to certain conditions, thus becoming a valuable tool for the prevention of disease and early detection of disease risks. Assessing salivary pH is relatively simple to do for research, but also in the dental practice. The information obtained can corroborate a patient’s oral status and be used to customize their treatment plan or give recommendations regarding eating habits and oral hygiene measures, highlighting the importance of periodic check-ups and preventive interventions.

Salivary pH was monitored with strips, which were not a pH meter. The electrode pH meter is the preferred method for pH evaluation due to its precision, but it poses disadvantages since it requires larger volumes of fluids and is more expensive [30]. pH strips offer several practical advantages that make them valuable in certain contexts. They are simple to use, require no specialized training, and provide rapid results, which is particularly useful for fieldwork, educational settings, or preliminary assessments. Additionally, they are cost-effective and portable, and do not require calibration or maintenance, unlike electrodes that need regular calibration and careful handling. Previously reported studies showed that strips are reliable for monitoring the pH of saliva and other biological fluids [31,32]. Also, of the available strips on the market, the version with multiple squares was used in this research, a type previously proven as being superior to the one with a single square per calibration [30].

The limitations of this research are as follows and should be considered by future studies. A larger sample size is desirable, considering the aspects for some outcomes, such as pH range, since according to our results, differences exist to a level very close to the *p*-value threshold. More information on participants’ characteristics with potential impact on resting salivary pH level and its variation, such as oral hygiene and oral health status, salivary flow rate, diet, and other baseline features, should be registered and assessed as confounders, especially in parallel group trials. Another limitation of this research is monitoring the salivary pH with strips, which is recommended in future studies to be used with a smaller measuring interval. The washout period of 40 min, accompanied by mouth rinsing with plain water, should be further investigated, since it is arguable in the context of the time needed for recovery of salivary pH, but it was found to be appropriate in our study for most of the subjects, posing the advantage of reducing loss to follow-up. According to our findings, the pattern variation for the two types of biscuits was similar, but they also registered some differences that are probably related to the differences between them, thus making it difficult to conclude which one had the biggest impact. In this regard, future studies should consider comparing products that have fewer and better-documented differences.

## 5. Conclusions

Results suggest the variation in salivary pH pattern after biscuit intake registers similarities, in general, but differences are noticeable between various product types. Cocoa biscuits with a sweet filling, when compared to oat biscuits, had a higher acidogenic effect and demonstrated a later start in an increasing trend as part of the recovery of salivary pH to its baseline. After the Oreo original intake, a mouth rinse with water promoted pH recovery. In order to have more accurate information on this clinically relevant topic, considering the high frequency of biscuit consumption during snacking and increased caries risk related to this habit, more high-quality studies, with methodology clarifications in terms of timepoints and duration of follow-up of salivary pH level after food intake, are required.

## Figures and Tables

**Figure 1 foods-14-04141-f001:**
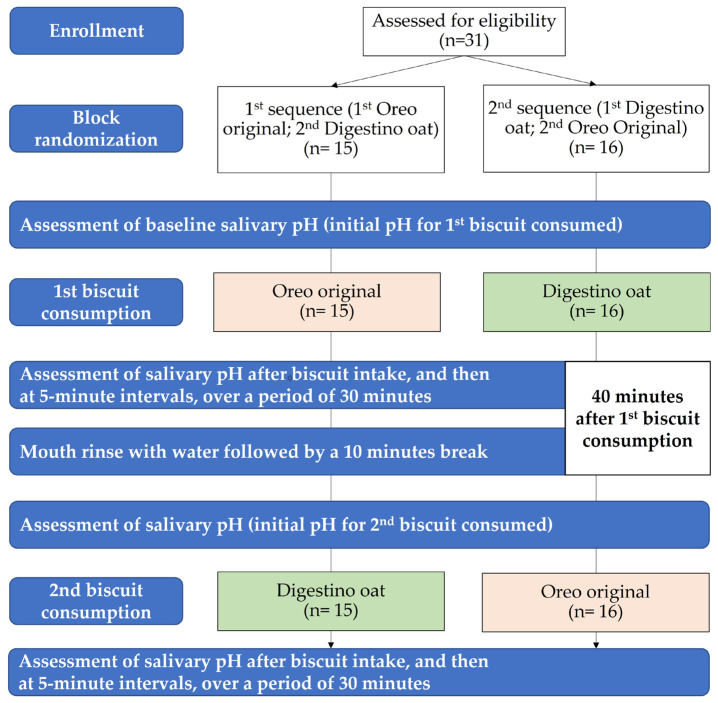
Flowchart of salivary pH assessment.

**Figure 2 foods-14-04141-f002:**
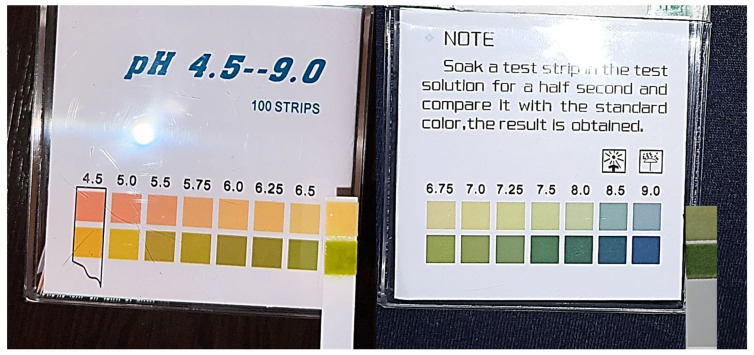
The strips used for the assessment of salivary pH.

**Figure 3 foods-14-04141-f003:**
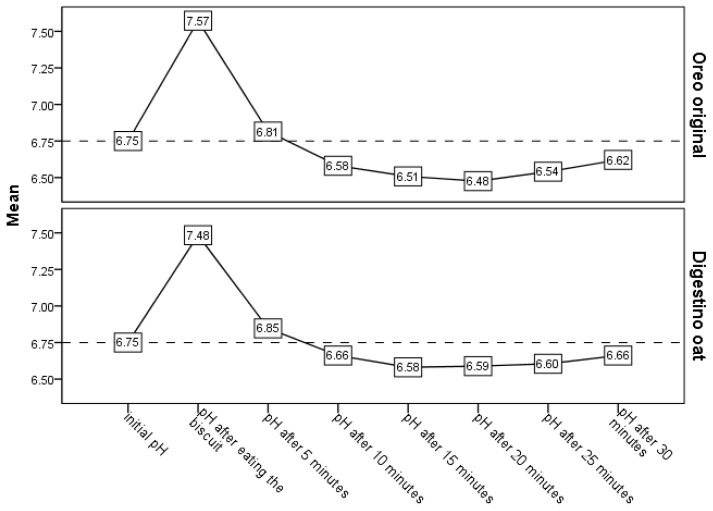
Variation in the salivary pH after intake of Oreo original and Digestino oat biscuits.

**Table 1 foods-14-04141-t001:** *p*-value of difference in initial salivary pH and salivary pH across timepoints used during follow-up for Oreo original and Digestino oat biscuits.

Initial Salivary pH vs. Salivary pH	*p* ^1^	
	Oreo Original	Digestino Oat
Right after consumption of the biscuit	0.001 *	<0.001 *
After 5 min	>0.999	>0.999
After 10 min	0.446	>0.999
After 15 min	0.009 *	0.446
After 20 min	0.001 *	0.446
After 25 min	0.044 *	>0.999
After 30 min	>0.999	>0.999

^1^ Friedman test, followed by pairwise comparisons with Bonferroni correction * Statistically significant difference (*p* < 0.05).

**Table 2 foods-14-04141-t002:** Comparative assessment of maximum pH, minimum pH, and pH range after intake of Oreo original vs. Digestino oat.

Variable	Oreo Original	Digestino Oat	*p* ^1^
Maximum pH (mean)	7.58	7.49	0.126
Minimum pH (mean)	6.38	6.44	0.149
pH range (mean)	1.20	1.06	0.051

^1^ Wilcoxon signed-rank test.

**Table 3 foods-14-04141-t003:** Correlation analysis of initial salivary pH with minimum pH, maximum pH, and pH range.

Correlation Analysis of Initial Salivary pH with	Oreo Original		Digestino Oat	
	r	*p*	r	*p*
Maximum pH	0.425	0.017 *	0.467	0.008 *
Minimum pH	0.780	<0.001 *	0.445	0.012 *
pH range	−0.296	0.106	0.166	0.374

Spearman test: r = correlation coefficient * Statistically significant correlations (*p* < 0.05).

**Table 4 foods-14-04141-t004:** Comparison of salivary pH in the following moments: (1) baseline; (2) 30 min after initial biscuit intake; (3) after mouth rinse and an extra 10 min follow-up.

Moment	Salivary pH					
	All Biscuits		Oreo Original		Digestino Oat	
	Mean	*p* ^1^	Mean	*p* ^1^	Mean	*p* ^1^
Baseline (1)	6.81	*p* < 0.001 * *p*_1–2_ = 0.007 * *p*_1–3_ = 0.170 *p*_2–3_ = 0.759	6.86	*p* = 0.001 * *p*_1–2_ = 0.008 * *p*_1–3_ = 0.513 *p*_2–3_ = 0.301	6.76	*p* = 0.012 * *p*_1–2_ = 0.555 *p*_1–3_ = 0.555 *p*_2–3_ > 0.999
At 30 min follow-up (2)	6.62		6.60		6.64	
At 40 min follow-up (after mouth rinse with water and an extra 10 min follow-up) (3)	6.69		6.73		6.64	

^1^ Friedman test, followed by pairwise comparisons with Bonferroni correction * Statistically significant difference (*p* < 0.05).

**Table 5 foods-14-04141-t005:** Variation in salivary pH from 30 min to 40 min follow-up.

		**Salivary pH Variation (pH at 40 min Minus pH at 30 min)**								
		**Oreo Original**					**Digestino Oat**			
		−0.25	0.00	0.25	0.50	Total	−0.25	0.00	0.25	Total
Salivary pH variation (pH at 30 min minus pH at baseline)	−1.00	0	0	1	0	1	0	0	0	0
	−0.50	0	0	2	**1**	3	0	2	1	3
	−0.25	0	3	**4**	0	7	1	1	0	2
	0.00	0	**3**	0	0	3	0	**11**	0	11
	0.25	**1**	0	0	0	1	0	0	0	0
Total		1	6	7	1	15	1	14	1	16

Bolded—those who recovered; Highlighted in gray—those with unchanged salivary pH from 30 to 40 min follow-up.

## Data Availability

The original contributions presented in this study are included in the article. Further inquiries can be directed to the corresponding authors.

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
