# Peer review of "Variation of Salivary pH After Sweet and Oat Biscuit Intake—A Crossover Randomized Controlled Trial"

_foods, 2025, doi:10.3390/foods14234141_

Round 1
Reviewer 1 Report
Comments and Suggestions for Authors
L51-56 Include the enzymes found in the oral cavity that are important for breaking down food.
In the introduction, include the problem statement more clearly.
L76 What is the importance of these two products? Attach information from different perspectives: production, consumption, ease or difficulty of chewing.
L90-96 Please include more information about the participants: average age, how they were selected or according to what international standard they were selected, did they smoke, how many were women and men, and what was the population size used for the experiment?
L98-100 How were these conditions established? It is suggested to include a flowchart that allows understanding of the entire methodological process applied.
Figure 1. Improve the image.
L123-125 Include statistical tests to define the use of the indicated statistical techniques. According to the results, these findings can be applied to the sensory field with what objective, and even correlated with instrumental techniques.
Include the limitations of the research prior to the conclusions.
Author Response
For research article “Variation of salivary pH after sweet and oat biscuit intake – a crossover randomized controlled trial”
|
Response to Reviewer 1 Comments |
|
Thank you very much for taking the time to review this manuscript. Please find the detailed responses below and the corresponding revisions and corrections highlighted in yellow in the re-submitted files. |
|
Point-by-point response to Comments and Suggestions for Authors |
|
Comments 1: L51-56 Include the enzymes found in the oral cavity that are important for breaking down food. |
|
Response 1: Thank you for pointing this out! We agree with this comment. Therefore, text was added in this regard in Introduction (in the 3rd paragraph, highlighted in yellow). |
|
Comments 2: L76 What is the importance of these two products? Attach information from different perspectives: production, consumption, ease or difficulty of chewing. |
|
Response 2: Thank you for this observation! We elaborated more on these products, text being found in Method section (4th paragraph), highlighted in yellow. Biscuits are frequently used during snaks. We elaborated on this topic in second paragraph of Discussion section. From commercial products available in our country, we selected 2 that by our knowledge was frequently consumed in our study sample, formed mainly by dental students. We also acknowledge when mentioning study limitations that there are multiple differences between them that we do not know, with possible impact on salivary pH, that should be better known and investigated in future researches |
|
Comments 3: L90-96 Please include more information about the participants: average age, how they were selected or according to what international standard they were selected, did they smoke, how many were women and men, and what was the population size used for the experiment? |
|
Response 3: Thank you for pointing this out! We mentioned sample characteristics, in regard to age and sex, in the beginning of the Result section. We also included some details upon general features, in the same paragraph, the first one of Results section. This data was collected by structured interviews, aspect mentioned now in Method section. Changes are highlighted in yellow. We added data on some general characteristics that were collected but were not included previously, as smoking status. We agree that a larger list would be relevant. And this is now mentioned in the end of Discussion, where study limitations are presented. Sample size was of 31 participants (mentioned in the beginning of Results section). We agree that our sample was not very big, but please let us argue that may be appropriate for a crossover RCT. Sample size of previous parallel group trials on the topic of salivary pH variation after biscuit consumption was reported as needed to be of at least 10 participants. By analyzing them, and other previous researches, on similar topics, we targeted to include at least 30 participants for this crossover trial. This information on sample size is now included in Method section (3rd paragraph, highlighted in yellow). Also, mentioning a larger sample size is recommended is mentioned in the study limitations, in the end of Discussion section. Changes were highlighted in yellow. Thank you for considering our arguments! |
|
Comments 4: L98-100 How were these conditions established? It is suggested to include a flowchart that allows understanding of the entire methodological process applied. |
|
Response 4: Thank you for this suggestion! We agree with the reviewer. Figure 1 was added corresponding to the flowchart. |
|
Comments 5: Figure 1. Improve the image. |
|
Response 5: Thank you for this suggestion! We change and hoe we improved the image. As adding a previous figure, mentioned Figure 1 is now figure 2. Hope this figure is better. |
|
Comments 6: L123-125 Include statistical tests to define the use of the indicated statistical techniques. According to the results, these findings can be applied to the sensory field with what objective, and even correlated with instrumental techniques. |
|
Response 6: Thank you for this suggestion! We added information that state the purpose of the statistical tests. Information added can be found in last paragraph of Method section and is now highlighted in yellow. Discussion upon usage of pH strips is added in a penultimate paragraph of Discussion section, and is mentioned as a study limitation. Changes are highlighted in yellow. |
|
Comments 7: Include the limitations of the research prior to the conclusions. |
|
Response 7: Thank you for this suggestion! This is an important part we missed. Information in this regard is now added as the last paragraph of Discussion section and highlighted in yellow. |
|
We hope that we have been able to address the issues that have been raised. We would like to thank you for your valuable suggestions and the time taken to consider our manuscript. We look forward to hearing from you. Best regards, The authors |

Reviewer 2 Report
Comments and Suggestions for Authors
I appreciate the opportunity to review the manuscript titled “Variation of salivary pH after sweet and oat biscuit intake – a crossover randomized controlled trial” by Cristina Teodora Preoteasa and co-authors.
The study compared salivary pH changes after chewing two commercially available biscuits (Oreo original — cocoa + creamy filling — vs. Digestino oat). Salivary pH was measured with pH strips before, immediately after, and then every 5 minutes for 30 min. Both biscuits produced a similar general pH time-course but Oreo induced a larger/longer acidogenic effect, Digestino recovered to baseline more often by 30 min, and a water mouth rinse improved pH recovery after Oreo.
Objectives
The objectives of the study are clearly stated.
Novelty
This is a controlled crossover study that compare specific commercial biscuit products using repeated short-interval pH sampling; this work adds practical data on recovery timing and the effect of a simple mouth rinse.
Methodology - Limitations of the study
- Washout period chosen to be 40 minutes. Please justify this.
- Did the authors record any information regarding the participants’ recent oral hygiene, salivary flow rate, or baseline diet prior to sessions?
- Sample size was n=31. Is this size usual for this kind of studies? Please elaborate.
- pH measurement method (pH strips) is of limited accuracy. Please discuss strip accuracy, and other limitations of this method in comparison to electrode method.
- Participants were instructed to “chew as much as possible”, shouldn’t this affect the result? Please elaborate.
Author Response
For research article “Variation of salivary pH after sweet and oat biscuit intake – a crossover randomized controlled trial”
|
Response to Reviewer 2 Comments |
|
Thank you very much for taking the time to review this manuscript. Please find the detailed responses below and the corresponding revisions and corrections highlighted in yellow in the re-submitted files. |
|
Point-by-point response to Comments and Suggestions for Authors |
|
Comments 1: I appreciate the opportunity to review the manuscript titled “Variation of salivary pH after sweet and oat biscuit intake – a crossover randomized controlled trial” by Cristina Teodora Preoteasa and co-authors. The study compared salivary pH changes after chewing two commercially available biscuits (Oreo original — cocoa + creamy filling — vs. Digestino oat). Salivary pH was measured with pH strips before, immediately after, and then every 5 minutes for 30 min. Both biscuits produced a similar general pH time-course but Oreo induced a larger/longer acidogenic effect, Digestino recovered to baseline more often by 30 min, and a water mouth rinse improved pH recovery after Oreo. Objectives The objectives of the study are clearly stated. Novelty This is a controlled crossover study that compare specific commercial biscuit products using repeated short-interval pH sampling; this work adds practical data on recovery timing and the effect of a simple mouth rinse. |
|
Response 1: Thank you very much, we appreciate your positive assessment |
|
Comments 2: Methodology - Limitations of the study
|
|
Response 2: We agree with the reviewer as this being a debatable aspect. Please consider our arguments. We choose this follow-up as after a pilot test, after which we found that after 30 minutes follow-up, mouth rinse with water and an extra 10 minutes brake salivary pH returned to its baseline. Also, several researches on the same topic, with the same and other eatables, found that pH recovers sooner than 30 minutes after consumption. This aspect is mentioned in Method section (6th paragraph), highlighted in yellow. This follow-up was chosen as was seen as being more appropriate, as diminishing the risk of loss to follow-up. Mouth rinse with water was also previously shown to positively contributing to returning the salivary pH to its resting value (aspect detailed in the penultimate paragraph of Discussion section). Considering the previous we choose to follow-up on salivary pH for 30 minutes, followed by mouth rinse with water, followed up by 10 minutes break, so resulting 40 minutes after consumption of the first biscuit. Even so, this being a debatable aspect, that need to be better known, was mention in the study limitations (end of discussion section), highlighted in yellow. Thank you for considering our arguments! |
|
Comments 3:
|
|
Response 3: We agree with the reviewer, this information is important Additionally, beside data on age and sex previously mentioned, data on some baseline characteristics were collected through structured interview. Now information on them is included in the beginning of Results section. Only part of the aspects mentioned was covered and only data from subjective evaluation was covered. Following your observation, this limitation of our research is now mentioned in the end of the discussion, highlighted in yellow. Also, mentioning that this data was collected through structured interview is mentioned in Method section, also highlighted in yellow |
|
Comments 4
|
|
Response 4: Thank you for pointing this out! Sample size was of 31 participants (mentioned in the beginning of Results section). We agree that our sample was not very big, but please let us argue that may be appropriate for a crossover RCT. Sample size of previous parallel group studies on the topic of salivary pH variation after biscuit consumption was reported as needed to be of at least 10 participants. By analyzing them, and other previous researches, on similar topics, we targeted to include at least 30 participants for this crossover trial. This information on sample size is now included in Method section, changes being highlighted in yellow (3rd paragraph of Method section). Also, mentioning a larger sample size is recommended is mentioned in the study limitations, in the end of Discussion section, highlighted in yellow. Thank you for considering our arguments! |
|
Comments 5:
|
|
Response 5: Thank you for this observation. We added explanations in the discussion section, found now as a penultimate paragraph. We provided some evidence that confirms that pH strips have advantages over pH meter in some situations (as fluids assessment) and results are considered accurate, even if current standard and best alternative is the pH meter. We also mentioned that using strips is a study limitation. Changes are highlighted in yellow. Thank you for considering our explanations and arguments! |
|
Comments 6:
|
|
Response 6: Thank you very much for this observation! We asked the subjects to chew bilaterally. We did that considering that mastication acts as a mechanical stimulus and by chewing bilaterally promotes saliva production on both sides of the mouth. Thank you for considering our explanations and arguments! |
|
We hope that we have been able to address the issues that have been raised. We would like to thank you for your valuable suggestions and the time taken to consider our manuscript. We look forward to hearing from you. Best regards, The authors |

Reviewer 3 Report
Comments and Suggestions for Authors
Comments on the paper
The study addresses a topic with limited data in the literature, but the novel contribution should be stated more explicitly in the Introduction and Conclusion. The current text focuses heavily on background information, without clearly defining what gap in knowledge this study fills.
Suggest clarifying:
- What has not been previously demonstrated in research on biscuit consumption and salivary pH.
- Whether this is the first controlled crossover study comparing chocolate/cream versus oat biscuits in adults.
- How the sample size of 31 was determined.
- Whether a power calculation was performed.
A short justification should be added, even if the sample is based on feasibility or pilot study criteria.
The demographic description is limited. Please consider adding:
- Oral health-related factors such as caries history, salivary flow rate (if available), dietary habits, or frequency of cariogenic food intake.
Even brief information would help contextualize the findings, as salivary pH responses can vary across individuals based on oral health status.
Salivary pH strips are practical but less precise than electrode-based digital measurement. It would strengthen the Methods section to address:
- Whether a calibration check of strips was performed.
- The degree of measurement resolution (e.g., accuracy to ±0.2 units).
- Whether the same operator read the color scale at all time points.
This is important because small but statistically significant differences are being reported.
it remains unclear whether the recorded pH values should be interpreted as unstimulated, stimulated, or a mixture of both. The Methods section states that “unstimulated salivary pH was done before and after each biscuit intake,” yet participants were required to chew the biscuits bilaterally before subsequent measurements. Because mastication is known to increase salivary flow rate and alter ion composition, the post-ingestion samples appear to represent stimulated saliva, whereas baseline samples are unstimulated.
This distinction is methodologically important because stimulated and unstimulated saliva differ in buffering capacity, flow rate, and biochemical composition, which may directly influence the magnitude and timing of pH changes. The manuscript would be strengthened by explicitly clarifying:
- Which measurements were unstimulated and which were stimulated,
- Whether stimulated and unstimulated samples were analysed together or separately,
- Whether flow rate was recorded, as differences in salivary output may contribute to measurement variability.
If flow rate was not recorded, this should be stated as a methodological limitation, as it is a major contributor to post-masticatory pH recovery.
The observed increase in salivary pH immediately after chewing is interesting and not consistent with much of the caries-related literature.
The authors offer a plausible explanation, but the Discussion could:
- Expand the interpretation,
- Consider the effect of stimulated saliva flow on bicarbonate concentration,
- Compare with classic Stephan curve physiology.
This would add depth to the discussion.
Findings on the effect of water rinsing are clinically valuable. However, the Discussion could be strengthened by:
- Clarifying whether the effect is mainly due to clearance of carbohydrate residue, dilution of acids, increased salivary flow, or a combination.
- Highlighting whether this could be recommended as a preventive behavior in daily life.
Minor Comments
- The Introduction is lengthy and could be shortened to make the rationale more focused and concise.
- Figure 2 would benefit from clearer labels and possibly confidence intervals or error bars.
- The term “follow-up” is used repeatedly for short-term measurements (within 40 minutes). Consider rephrasing as “observation period” to avoid confusion with clinical follow-up terminology.
- Some grammatical and typographical errors are present; a careful language edit would improve readability.
To strengthen the methodological robustness, the authors should clarify:
- The smallest pH resolution of the strips (e.g., 0.5 increments or finer),
- Whether calibration or quality control procedures were performed,
- Whether multiple observers were used to assess reading reliability, and
- Whether these strips have previously been validated in peer-reviewed research for saliva.
Comments on the Quality of English Language
Needs improvment
Author Response
For research article “Variation of salivary pH after sweet and oat biscuit intake – a crossover randomized controlled trial”
|
Response to Reviewer 3 Comments |
|
Thank you very much for taking the time to review this manuscript. Please find the detailed responses below and the corresponding revisions and corrections highlighted in yellow in the re-submitted files. |
|
Point-by-point response to Comments and Suggestions for Authors |
|
Comments 1: The study addresses a topic with limited data in the literature, but the novel contribution should be stated more explicitly in the Introduction and Conclusion. The current text focuses heavily on background information, without clearly defining what gap in knowledge this study fills. Suggest clarifying: What has not been previously demonstrated in research on biscuit consumption and salivary pH. Whether this is the first controlled crossover study comparing chocolate/cream versus oat biscuits in adults. |
|
Response 1: Thank you for pointing this out! We agree with this comment. Novel contribution is now highlighted in the Introduction section (penultimate paragraph) and text had been added to Conclusions section, changes being highlighted in yellow. |
|
Comments 2: How the sample size of 31 was determined. Whether a power calculation was performed. A short justification should be added, even if the sample is based on feasibility or pilot study criteria. |
|
Response 2: We agree with the reviewer. We have, accordingly, added text explaining why we targeted inclusion of at least 30 participants. Sample size of previous parallel group studies on the topic of salivary pH variation after biscuit consumption was reported as needed to be of at least 10 participants. By analyzing them, and other previous researches, on similar topics (pH variation after intake of different eatables), we targeted to include at least 30 participants for this crossover trial. This information on sample size is now included in Method section (3rd paragraph of it). Also, mentioning a larger sample size is recommended in future researches is now mentioned in the study limitations, in the end of Discussion section. All changes are highlighted in yellow. Thank you for considering our arguments! |
|
Comments 3 The demographic description is limited. Please consider adding:
Even brief information would help contextualize the findings, as salivary pH responses can vary across individuals based on oral health status. |
|
Response 3: We agree with the reviewer, this information is important. Additionally, beside data on age and sex previously mentioned, data on some baseline characteristics were collected through structured interview. Now information on them is included in the beginning of Results section (1st paragraph). Only part of the aspects mentioned was covered, and we agree that an extensive list would be better. Following your observation, this limitation of our research is now mentioned in the end of the Discussion, in the last paragraph of the section. Also, mentioning that this data was collected through structured interview is mentioned in Method section. All changes are highlighted in yellow. |
|
Comments 4 Salivary pH strips are practical but less precise than electrode-based digital measurement. It would strengthen the Methods section to address:
This is important because small but statistically significant differences are being reported. |
|
Response 4: Thank you for this comment! We agree that electrode base digital measurements would be more precise, but we considered, as previous researches showed that pH strips were appropriate (and actually this is what was available for us). We argue that in the current version of the manuscript, in the penultimate paragraph of Discussion section. We also mentioned this limitation when stating study limitations in the last paragraph of Discussion section. In Method section now is mentioned that measurements interval used can be acknowledged in Figure 2 (priorly Figure 1) and that one person (ACB) performed all pH recordings. All changes are highlighted in yellow. |
|
Comments 5 it remains unclear whether the recorded pH values should be interpreted as unstimulated, stimulated, or a mixture of both. The Methods section states that “unstimulated salivary pH was done before and after each biscuit intake,” yet participants were required to chew the biscuits bilaterally before subsequent measurements. Because mastication is known to increase salivary flow rate and alter ion composition, the post-ingestion samples appear to represent stimulated saliva, whereas baseline samples are unstimulated. This distinction is methodologically important because stimulated and unstimulated saliva differ in buffering capacity, flow rate, and biochemical composition, which may directly influence the magnitude and timing of pH changes. The manuscript would be strengthened by explicitly clarifying:
If flow rate was not recorded, this should be stated as a methodological limitation, as it is a major contributor to post-masticatory pH recovery. |
|
Response 5: Thank you very much for this comment! Indeed, we made an error when we wrote. We corrected that, in Method (6th paragraph). We also acknowledge the importance of flow-rate, and not recording it is now mentioned in study limitations, in the end of Discussion section. All changes are highlighted in yellow. Thank you again for pointing that out and allowing us to correct the mistake! |
|
Comments 6 The observed increase in salivary pH immediately after chewing is interesting and not consistent with much of the caries-related literature.
This would add depth to the discussion. |
|
Response 6: Thank you for your comments and your suggestions! There was another research, reported by Pachori et al that found a similar increase of salivary pH after solid food consumption, but not after liquid ones. Was the only reported research identified by us, similar in this regard. We explained that aspect by chewing acting as a stimuli for saliva production, and we added arguments suggested by you. These are now included in the 3rd paragraph of Discussion section, highlighted in yellow. Thank you again! |
|
Comments 7 Findings on the effect of water rinsing are clinically valuable. However, the Discussion could be strengthened by:
|
|
Response 7: Thank you for your comment! 7th paragraph in Discussion section, addressing these results, was reformulated and your valuable suggestions were added. |
|
Comments 8: Minor Comments
|
|
Response 8: We agree with the reviewer, we think also the introduction in a somehow lengthy, but please consider our opinion that it includes important and relevant aspects that shape the context of our research. Thank you! |
|
Comments 9:
|
|
Response 9: We did one of the variants you suggested, which is inserted below. We prefer the simpler version as from our point of view it does not bring additional very important information. Thank you!
|
|
Comments 10:
|
|
Response 10: Thank you very much for this suggestion! We considered follow-up as being a length of time the person will be monitored. We consider that in the context of prospective studies this term is frequently used and well understood. But we are not native English speakers, and if this is a mistake, we will correct it. Thank you for considering our opinion! |
|
Comments 11:
|
|
Response 11: We agree with the reviewer. Corrections were made in the current form of the manuscript. Hope we improved our manuscript. Thank you! |
|
Comments 12: To strengthen the methodological robustness, the authors should clarify:
|
|
Response 12: Thank you very much for these suggestions! Data on pH intervals registered with the pH strips used are mentioned to be observed in Figure 2 (Method section), and we mentioned this aspect when stating study limitations (last paragraph of Discussion), aspects reflecting these changes from previous manuscript being highlighted in the current version in yellow. We cannot say calibration or quality control procedures were performed, but before establishing the final version of the method we did some test. Among other aspects, we compared 3 commercial products of strips. Among them there were two with multiple squares, as the one presented, and they gave same results. But we did not record that data, were part of our pilot testing, which helped us to establish the method features for this research. Only one person (ACB) read all values related to pH recordings, aspect that is now mentioned in the Method section (7th paragraph) and is now highlighted in yellow. By our knowledge and previous researches, pH strips are considered accurate for fluids evaluation. Information in this regard was added and placed as the penultimate paragraph of Discussion section, but considering that pH meter is more precise and preferred, this aspect was mentioned as a study limitation. Changes are highlighted in yellow. |
|
Response to Comments on the Quality of English Language |
|
We regret there were problems with the English. The paper has been carefully revised to improve the grammar and readability. Hope we improved our manuscript. Thank you! |
|
We hope that we have been able to address the issues that have been raised. We would like to thank you for your valuable suggestions and the time taken to consider our manuscript. We look forward to hearing from you. Best regards, The authors |

Round 2
Reviewer 1 Report
Comments and Suggestions for Authors
Thank you for your responses.
Reviewer 2 Report
Comments and Suggestions for Authors
The authors addressed my previous comments. No further comment.